# Structural Equation Modelling for Predicting the Relative Contribution of Each Component in the Metabolic Syndrome Status Change

**DOI:** 10.3390/ijerph19063384

**Published:** 2022-03-13

**Authors:** José E. Teixeira, José A. Bragada, João P. Bragada, Joana P. Coelho, Isabel G. Pinto, Luís P. Reis, Paula O. Fernandes, Jorge E. Morais, Pedro M. Magalhães

**Affiliations:** 1Research Centre in Sports Sciences, Health and Human Development (CIDESD), 5001-801 Vila Real, Portugal; jbragada@ipb.pt (J.A.B.); morais.jorgestrela@gmail.com (J.E.M.); 2Department of Sport Sciences, Instituto Politécnico de Bragança (IPB), 5300-253 Bragança, Portugal; pmaga@ipb.pt; 3North East Local Health Unit (ULSNE)—Health Care Unit of Santa Maria, 5301-852 Bragança, Portugal; joao.bragada@ulsne.min-saude.pt (J.P.B.); joana.coelho@ulsne.min-saude.pt (J.P.C.); isabel.pinto@ulsne.min-saude.pt (I.G.P.); luis.reis@ulsne.min-saude.pt (L.P.R.); 4Applied Management Research Unit (UNIAG), Instituto Politécnico de Bragança (IPB), 5300-253 Bragança, Portugal; pof@ipb.pt

**Keywords:** metabolic syndrome, multilevel modelling, prediction, progression, public health

## Abstract

Understanding the factor weighting in the development of metabolic syndrome (MetS) may help to predict the progression for cardiovascular and metabolic diseases. Thus, the aim of this study was to develop a confirmatory model to describe and explain the direct and indirect effect of each component in MetS status change. A total of 3581 individuals diagnosed with MetS, aged 18–102 years, were selected between January 2019 and December 2020 from a community-representative sample of Portuguese adults in a north-eastern Portuguese region to test the model’s goodness of fit. A structural equation modelling (SEM) approach and a two-way ANOVA (age × body composition) were performed to compare the relative contribution of each MetS component using joint interim statement (JIS). Waist circumference (β = 0.189–0.373, *p* < 0.001), fasting glucose (β = 0.168–0.199, *p* < 0.001) and systolic blood pressure (β = 0.140–0.162, *p* < 0.001) had the highest direct effect on the change in MetS status in the overall population and concerning both sexes. Moreover, diastolic blood pressure (DBP), triglycerides (TG) and high-density lipoprotein cholesterol (HDL-c) had a low or non-significant effect. Additionally, an indirect effect was reported for age and body composition involving the change in MetS status. The findings may suggest that other components with higher specificity and sensitivity should be considered to empirically validate the harmonised definition of MetS. Current research provides the first multivariate model for predicting the relative contribution of each component in the MetS status change, specifically in Portuguese adults.

## 1. Introduction

Metabolic syndrome (MetS) is a public health concern defined by a cluster of major risk factors, such as central obesity, dysglycemia, dyslipidemia and hypertension [1]. For several years, studies have reported that the combination of these factors increases the risk of atherosclerotic cardiovascular diseases (CVD), type 2 diabetes mellitus (T2DM), neurological disorders and cancers [2,3,4]. The prevalence of MetS has been increasing worldwide over the last decades due to obesity-related factors, such as a sedentary lifestyle and an unhealthy diet [5,6]. Southern European countries showed a higher prevalence of MetS in comparison to the remaining European countries (i.e., Italy, Spain, and Portugal) [7]. Specifically, in the Portuguese context, a high prevalence of MetS has also been reported among 37.2% to 54.51% of the total population [7,8,9]. Additionally, a high prevalence has been described for MetS components when analysed separately such as: overweight (39.1%) and obesity (28.6%) [10], hypertension (42.2%) [11], high-risk lipid profile (73.0%) [12,13] and high levels of insulin resistance (41.6%) [14,15]. Furthermore, physical inactivity, a hypercaloric diet and a sedentary lifestyle have been also highly reported in Portuguese population [16,17]. In Portugal, 46% of the population is physically inactive (i.e., 40% for men and 52% for women) [18]. Subsequently, the lack of the protective regular exercise physical activity benefits causes an increase in metabolic disorders [19]. In addition, MetS is highly age-dependent, wherefore the ageing of the Portuguese population also justifies an in-depth examination of the MetS change [19,20].

Several clinical guidelines have been developed for the diagnosis of MetS, including those from the World Health Organization [20], European Group for the Study of Insulin Resistance (EGIR) [21], National Cholesterol Education Program Adult Treatment Panel III (NCEP-ATP III) [22], American Association of Clinical Endocrinologists (AACE) [23], International Diabetes Federation (IDF) [24] and American Heart Association/National Heart, Lung and Blood Institute (AHA/NHLBI) [25]. Nevertheless, these different definitions of MetS have led to some confusion in the literature. This occurred due to the use of different clinical indicators and biological cut-offs, demonstrating that some MetS components can be overestimated in relation to others [26,27]. Upon that, joint interim statement (JIS) criteria defined MetS diagnosis by the presence of an elevated waist circumference (WC), low high-density lipoprotein cholesterol (HDL-c), elevated triglyceride levels (TG), impaired fasting glucose (IFG), elevated systolic (SBP) and/or diastolic blood pressure (DBP) [28]. MetS diagnosis is confirmed when three of these five components are present. However, the definition does not assign different weights to each MetS component [27]. 

Until now, the relative contribution of each component in MetS status has been mainly described resorting to classical statistical analysis (e.g., analysis of variance and regression models) [29]. However, multilevel structural equation modelling (SEM) is a more helpful procedure in gathering insight about the most critical MetS components [30,31,32,33]. It is actually more important considering that the cardio-metabolic morbidity and mortality have a 2.5-fold linear increase with the number of MetS components [34]. The literature reported that SEM allowed for a more realistic assumption on measurement errors, factor loadings and better model fit indices comparing with other multilevel analyses [35,36,37]. SEM techniques can be applied to estimate the pathways for each MetS-component, considering the aforementioned definitions as a theoretical model [38]. Studies have pointed out the relationship between MetS and independent predictors, such as sociodemographic factors [30], modifiable risk factors [31], cardio-metabolic risk factors [32], neurophysiological factors and genetic determinants [33]. Nevertheless, SEM analysis has been applied to estimate the factor weighting in the diagnosis of MetS, determined by the presence or absence, neglecting the relative contribution of each component in the MetS status change [39]. Previous studies have validated confirmatory models for MetS score in children, middle-age and older adults, providing the differential contribution for each component in the diagnosis of MetS [40,41,42,43,44,45]. On this basis, cardiometabolic risk can be predicated by MetS components using standardised scores [43,45], latent factors [41] or indexes [42,44]. However, there is a weak description of a confirmatory model to explain direct and indirect effect of each component in the change in MetS status [19,40]. Understanding the relative contribution of each component in the development of MetS may help predict the progression of the syndrome to cardiovascular and metabolic diseases [45]. Additionally, usually, published studies are conducted on several specific populations, nations or regions (i.e., United Kingdom, China, Iran, United States, Finland, Spain) [33,38]. 

However, and to the best of our knowledge, confirmatory models for predicting the change in MetS status in the Portuguese population have not yet been developed. Thus, the aim of this study was to develop a confirmatory model, through SEM analysis, to describe and explain the direct and indirect effect of each component in MetS status change, specifically in a community representative sample of Portuguese adults from a north-eastern Portuguese region. It was hypothesised that WC, SBP and IFG achieved a higher relative contribution for predicting the MetS status change. In addition, it is hypothesised that the relative contribution of each component in the change in MetS status differs by sex.

## 2. Materials and Methods

### 2.1. Study Design and Population

A total of 3581 individuals aged 18–102 years, among which 1914 were women (66.73 ± 12.89 years) and 1667 were men (mean age of 64.08 ± 14.23 years), were included in the sample. A cross-sectional, observational and retrospective analysis was conducted between January 2019 and December 2020 from patients’ clinical records of two primary health care centres in a north-eastern Portuguese region. A total of 18,890 participants were analysed, among which 12,320 participants were excluded from the data analysis considering the following exclusion criteria: (i) participants with <18 years; (ii) missing information about MetS clinical criteria’s, height, weight, BMI and demographic considerations. From those, 2989 individuals did not have MetS diagnosis (i.e., less than or equal to 2 components). All sampled individuals were diagnosed with MetS. Subjects were divided into three age-groups following standard recommendations [46]: young adults (18–39 years), middle-aged adults (40–64 years) and older adults (≥65 years). Table 1 presents the descriptive statistics for age groups, BMI and MetS components according to sex. Additionally, the distribution and frequencies for 3-, 4- and 5-MetS components was presented by sex. A normal distribution and a variance homogeneity were reported for all the MetS-components.

### 2.2. Data Collection

#### 2.2.1. Anthropometric Measures

Anthropometric measures were evaluated during clinical practice using standard guideless by the International Society for the Advancement of Kinanthropometry (ISAK) [47]. The measurements were carried out by an expert evaluator in each health centre. Body mass, height and WC were recorded by the average of three measurements to the nearest 0.1 International Units (IU). Body mass (kg) was evaluated using an electronic scale Tanita MC 780-P MA^®^ (Tanita Corporation, Tokyo, Japan) with minimum clothing. Height (cm) was collected using an electronic stadiometer (Seca, Hamburg, Germany). Waist circumference (cm) was measured using a flexible steel tape at the midpoint among the top of the iliac crest and the lower margin of the lower palpable rib [48]. The evaluator performed the measurement three times, and the average was used for further analysis. The IntraClass Correlation Coefficient (ICC) revealed an almost perfect agreement (ICC = 0.999). Body max index (BMI) was calculated by dividing weight by the square of height (m^2^). European cut-offs were used to define overweight (25.0 to 29.9 kg/m^2^) and obesity (≥30 kg/m^2^) [49]. 

#### 2.2.2. Laboratory Analysis and Blood Pressure

All subjects had at least one record of blood tests in their clinical process, if valid for at least 6 months. Blood samples were collected from a collaborative laboratory at the two primary health centres after 8/10 h of overnight fasting, using standard laboratory procedures and routine enzyme methods [50]. HDL-c, TG and fasting glucose (FG) were directly measured. Blood pressure (mmHg) were assessed using a standard protocol, from which SBP and DBP were measured three times in seating position with 1 min interval between measurements (the means of the three measurements were considered for the analysis).

### 2.3. MetS Definition

MetS was defined in this study using the JIS criteria [28]. MetS diagnosis is confirmed when three of the following five components are present: elevated WC (i.e., population- and country-specific delimitations), elevated TG (i.e., ≥150 mg/dL or 1.7 mmol/L), reduced HDL-c (i.e., <40 mg/dL or 1.0 mmol/L in males; <50 mg/dL or 1.3 mmol/L in females), elevated SBP (i.e., ≥130 mmHg), elevated DBP (i.e., ≥85 mmHg) and FG (i.e., ≥100 mg/dL or 5.6 mmol/L). European cut off points were considered for the WC measurements, specifically: WC ≥ 88 cm in women and WC ≥ 102 cm in men. Additionally, drug treatment for each MetS component was considered as an alternative indicator for the diagnosis of the syndrome. 

### 2.4. Theoretical Model

The theoretical model was designed based on JIS criteria for MetS (Figure 1), reporting the direct effect for each MetS component and the indirect effect for age and BMI [30]. A quasi-linear increase across age was verified in the MetS prevalence for both sexes with a decline from the eighties onwards. Additionally, sex disparity in MetS prevalence is well-documented in Portuguese population [7,8]. The theoretical model was stratified into two levels: (i) the effect of age and body composition (i.e., BMI) in each MetS-component; (ii) the weighting of each MetS-Component in the diagnosis of MetS. 

### 2.5. Statistical Analysis

Descriptive statistics (means and standard deviations) were performed for all the analysed variables with 95% confidences intervals (CI). The Kolmogorov–Smirnov and Levene’s test were used to assess the normality and homogeneity, respectively. A two-way ANOVA (age × BMI) was used to compare continuous independent variables (i.e., MetS components) amongst the overall population and both sexes. The effect size eta square (η^2^) was computed and interpreted as: (i) without effect if 0 < η^2^ ≤ 0.04; (ii) minimum if 0.04 < η^2^ ≤ 0.25; (iii) moderate if 0.25 < η^2^ ≤ 0.64; and (iv) strong if η^2^ > 0.64. When a significant difference occurred, Turkey’s post-hoc tests were used to identify localised effects. Statistical significance was set at *p* ≤ 0.05 [51,52]. A structural equation modelling (SEM) was performed using a path-flow method with a two-step maximum likelihood approach [53,54]. An adjusted goodness-of-fit model was obtained based on previous reports [35,36,37]: (a) the independent variables were inserted inside squares; (b) the links were represented by an arrow between two variables, with one variable determining the other; (c) beta values (β) report the contribution of one variable to the other. That is, when the exogenous variable increases (i.e., MetS components) by one unit the endogenous variable increases (i.e., MetS diagnosis) by the amount of the beta value; (d) residual errors (e) and determination coefficient (β) represents the variable predictive error and the variable predictive value, respectively; and (f) age and BMI represent the moderate variables for the diagnosis of MetS diagnosis, also being the exogenous variables for MetS components. 

A confirmatory model was obtained from the theoretical model confirmation [36]. Afterwards, a path-flow analysis was performed by the estimation of linear regression standardised coefficients between the dependent and independent variables [35,36]. All assumptions to perform the path-flow analysis were considered. Whenever appropriate, simple or multiple linear regression analysis was computed according to the theoretical model. Standardised regression coefficients (β) were considered [35,37]. The effect size of the disturbance term, reflecting unmeasured variables, for a given endogenous variable, was 1–R^2^. The significance of each β was assessed through the Student’s *t*-test [36]. To measure the model’s goodness-of-fit, the standardised root mean square residuals (***SRMR***) was calculated:(1)$$SRMR=\sqrt{\frac{\sum_{i=1}^{p}\sum_{i=1}^{q}{\left(rij-pij\right)}^{2}}{p+q}}$$
where ***r*** is the Pearson correlation coefficients and ***p*** the correlation predicted by the model (based on total effect), the addiction of the direct and indirect effects plus spurious effects was calculated by: (a) ***r*** is the Pearson correlation coefficients; and (b) ***p*** the correlation predicted by the model [36,37]. The ***SRMR*** measures the standardised difference between the observed covariance and the predicted covariance. It is considered a rule of thumb that if: (a) ***SRMR*** < 0.1, the model adjusts to the theory; (b) ***SRMR*** < 0.05, the model adjusts very well to the theory; and (c) ***SRMR*** ~ 0, the model is perfect [35]. All statistical analysis was conducted using SPSS for Windows Version 22.0, IBM SPSS AMOS 23.0 (SPSS Inc., Chicago, IL, USA) and JASP software (JASP Team, Jasper, IN, USA, 2019).

## 3. Results

### 3.1. Descriptive and Comparison Analysis

Table 2 showed the mean comparison between the overall population, women and men in each MetS component according to age groups, BMI bands and interaction effect amongst both independent factors (i.e., for age group × BMI band). When considering the age group, all MetS components presented significant differences in women (*F* = 9.001–122.625; η^2^ = 0.000–0.019, *p* < 0.05 to *p* < 0.001), men (*F* = 5.088–199.313; η^2^ = 0.003–0.100, *p* < 0.05 to *p* < 0.001) and the overall population (*F* = 14.066–75.807; η^2^ = 0.000–0.008; *p* < 0.05 to *p* < 0.001). When considered BMI bands, women presented significant differences for FG (*F* = 14.067; η^2^ = 0.015; *p* < 0.001), HDL (*F* = 16.37; η^2^ = 0.017; *p* < 0.001), WC (*F* = 262.754; η^2^ = 0.216; *p* < 0.001). Men showed significant differences in BMI bands for TG (*F* = 3.785; η^2^ = 0.005; *p =* 0.023), HDL (*F* = 6.109; η^2^ = 0.007; *p =* 0.002) and WC (*F* = 408.137; η^2^ = 0.330; *p <* 0.001). The overall population had significant differences in BMI bands for FG (*F* = 5.945; η^2^ = 0.003; *p* = 0.003), TG (*F* = 5.412; η^2^ = 0.003; *p* = 0.003), HDL (*F* = 27.302; η^2^ = 0.015; *p <* 0.001) and WC (*F* = 610.302; η^2^ = 0.255; *p <* 0.001). Interaction effects amongst age groups x BMI bands were only founded for women in TG (*F* = 2.890; η^2^ = 0.006; *p =* 0.021). 

### 3.2. Structural Equation Model

Using a structural equation model (SEM), it was noted that WC had the highest direct effect on MetS diagnosis and evolution (β = 0.189 to 0.373, *p* < 0.001) in the three sub-samples (i.e., overall population, women and men). Afterwards, FG (β = 0.168 to 0.199, *p* < 0.001) and SBP (β = 0.140 to 0.162, *p* < 0.001) were the MetS-components with the highest effect. Otherwise, DBP had the lowest effect for MetS with available significance for the overall population (β = −0.063, *p* < 0.001) and women (β = −0.072, *p* < 0.001). TG only showed significant effect on men (β = 0.048, *p* < 0.001).

At the first level, age and BMI had a direct effect on all MetS-components, except for age → HDL in the overall population and men, age → TG in women, age BMI → SBP for the overall population and women. Age had a direct effect on BMI in three sub-samples (β = −0.177 to −0.184, *p* < 0.001). A goodness-of-fit model (good adjustment—***SRMR*** < 0.05) was performed for the overall population (Figure 2A), women (Figure 2B) and men (Figure 2C).

## 4. Discussion

The aim of this study was to develop a confirmatory model, by SEM analysis, to describe and explain the direct and indirect effect of each MetS component in Portuguese adults from a north-eastern Portuguese region. Therefore, the relative contribution of each component was identified for predicting the MetS status change. As hypothesised, the MetS components with the greatest contribution in the changes of the syndrome were WC, FG and SBP (Figure 2). The current findings were congruent with the differences between age groups, BMI bands and MetS-components, reporting an effect of age and body composition in all MetS components, except for the links between age → HDL in the overall population and men, age → TG in women, and age BMI → SBP in the overall population and women (Table 2). Another confirmed hypothesis in this SEM analysis is the sex disparity in the relative contribution of each component for the change in MetS status. 

The harmonised definition of MetS, proposed by Alberti et al. [28], continues to be extensively analysed using multivariate approaches in order to assess the weight of each MetS component [38]. However, regardless of the methodological procedures adopted, the literature seems to clearly define abdominal obesity and insulin resistance as the most influential variables in MetS expression [30,38]. Current research adds further evidence in this direction, reporting a highest direct effect of abdominal obesity and fasting glucose on 3-, 4- and 5-MetS components, representing the most critical factors for the changes in the syndrome status, concerning this specific community of the Portuguese population (Figure 2). Conceptually, excess visceral adiposity tends to trigger the development of MetS, explaining why hyperinsulinemia may not be associated with an increase in fasting or postprandial glucose for many years [55,56]. In more genetically predisposed individuals, or in those with environmental and behavioural factors such as a sedentary lifestyle and nutrition, there is a gradual glucose overload and a consequent development of impaired insulin secretion and inadequate insulin action [45,57]. This pathogenic mechanism explains the relative or absolute deficiency in insulin secretion by the pancreatic Beta cells and/or a greater or less resistance to this hormone by the cells of the target tissue [55]. Additionally, it is difficult to measure the hyperinsulinemia in clinical practice, by the analytic parameters of the fasting and postprandial glucose, and the inflammatory and pathologic process often turns out to be silent, which can trigger an increased risk of cardiovascular and metabolic diseases [2,3,4]. 

On the other hand, the adipose tissue dysfunction alters the significant contribution of adipocyte-derived hormones or cytokines expression (i.e., leptin, adiponectin, interleukin-6) in vital immunological, cardiovascular, metabolic and endocrine functions, linking obesity and impaired insulin sensitivity [58]. Hence, these physiopathological mechanisms are expressed in obesity-associated and metabolic defects in other tissues, such as atherogenic dyslipidemia, endothelial dysfunction and hypertension [59]. In this complex process, apart from obesity-associated insulin resistance, other metabolic factors play a key role in the evolution of MetS, including inflammatory factors [60], defects in the target cell (i.e., receiver and post-receiver) [2], and an increase in insulin counterregulatory hormones and anti-insulin antibodies (AIA) [61]. Consequently, endothelial dysfunction and dyslipidemia leads to atherosclerotic mechanisms and arterial intima-media thickness [62,63], and, consequently, to an increased risk of micro and macrovascular complications in chronic disease, with a particular emphasis on T2DM [2], which explains the inclusion of SBP, DBP, TG and HDL in the harmonised definition of MetS [28]. Hypertension tends to be associated with metabolic risk factors, and about half of hypertensive patients are insulin-resistant [11,63]. Dyslipidemia includes elevated levels of fatty acids, apolipoprotein B (ApoB), TG, high levels of low-density lipoproteins (LDL) and low levels of HDL, leading to an increase in CVD risk [62]. The current findings confirmed the direct effects of systolic hypertension in the changes of MetS status; however, the remaining indicators did not show such a sustained effect in the interest groups. Based on this SEM analysis, TG only showed significant effects on men and any relative contribution is related to HDL in MetS evolution for sampled population (Figure 2). Jiang et al. [29] also established a distinct age-related prevalence of MetS with a sex disparity. The population-based cohort studies have reported a higher prevalence of MetS in women but a higher prevalence of MetS in some specific populations [5,6,7,8]. Thereby, the prevalence of MetS seems to have a wide variation between sexes across different populations, ethnicities and nationalities [9,49].

The adjusted goodness-of-fit model obtained in SEM analysis demonstrates some inconsistencies with the theoretical model based on the harmonised definition (Figure 1). Indeed, the pathogenesis of MetS remains unclear, and several other components have been shown to have higher sensitivity and specificity for predicting MetS [27,64]. Nonetheless, the harmonised definition seems to have left out some important adjustments to the variables that report on each MetS component such as the waist-to-height ratio [64], waist-to-hip ratio [65], mean arterial pressure [66] and LDL levels [67], all of which have been reported as better screening criteria for MetS. BMI has been previously considered in AACE and IDF definitions and retains its usefulness as an alternative parameter for the assessment of central obesity [23,24]. However, the current multivariate model did not report effects for SBP in the overall population or in women (Figure 2). Furthermore, the interactive effects between age and BMI (Table 2), in congruence with the indirect age effects in MetS (Figure 2), reported by the multivariate model, demonstrates the role of functional senescence in the progressive decline associated with age in metabolic activity and function [50]. In this study, BMI and HDL were inversely related with a meaningful relationship (Figure 2). Indeed, improving the lipid profile is highly dependent on reducing body fat and lifestyle interventions to better control the CVD risk factors associated with MetS progression [13,29,61]. Physical activity and exercise seem to play a key role in mitigating adverse ageing effects, as well as physiopathological mechanisms reported for MetS [68,69]. Implementing effective and evidence-based exercise strategies for prevention, initial weight loss, and maintenance of weight loss are, therefore, crucial [70]. However, it has been demonstrated that for the treatment of obesity, this amount of physical exercise is not enough [71]. Individualised prescription of physical exercise and an improved lifestyle and nutrition seem to promote better results in the reduction in MetS-associated cardiovascular risks factors [72,73]. Overall, exercise modes decrease MetS clinical biomarkers. Nevertheless, aerobic training seems to produce the greatest outcomes, especially regarding high-intensity training [55,72,74]. Chronic adaptations linked to an optimised glycaemic control, fat mass reduction and cardiovascular fitness improvement are normally associated with aerobic training [60]. Moreover, resistance exercise seems to have a positive effect on MetS prevention when combined with aerobic exercise [74]. Specifically, the benefits of aerobic exercise include an increased expression of glucose transporter (GLUT4) proteins in skeletal muscle cells [75], an improvement of the vasodilation function mediated by bioequivalent nitric oxide (${\mathrm{NO}}_{3}^{-}$) [76], and a reduction of the hepatic glucose production as due to the hormonal regulation [77]. Additionally, a better blood lipid profile increases the lipoproteins lipase activity as ApoB in muscle capillaries and, subsequently, promotes a concomitant oxidation of free fatty acids (FFA) in trained muscle cells, by diminishing the ectopic lipid deposition and lipotoxicity [78]. Likewise, adipokines such as leptin, adiponectin, and lipocalin-2 are involved in the regulation of autoimmunity, leading to the modulation of the immune system, MetS and arthritic diseases [79]. Thus, future research should add independent variables into the multivariate model to prescribe and control the exercise intensity and volume on the basis of reference values for prescribing exercise therapy in MetS [31,80,81]. This could allow measuring optimal exercise intensity and volume to improve insulin effectiveness and promote related cardiometabolic gains along with body weight control [69]. Furthermore, controlling the impact of socio-demographic, nutritional or behavioural variables associated with modifiable risk factors is another point to consider in future models for predicting MetS status change [5,30,73]. 

Retrospective data were collected from patients’ clinical records of two primary health care centres in a Portuguese NUTS (i.e., nomenclature of territorial units for statistics). There are some limitations to this study. In fact, inter-regional differences should be considered when applying this model, as well as the restrictions inherent to a retrospective observational analysis [7,8,9]. Additionally, assessing the insulin effectiveness would require other diabetes-related parameters, such as the 2 h oral glucose tolerance test (11.1 mmol/L) (OGTT) by 2 h plasma glucose (2hPG) and/or haemoglobin A1c (HbA1c) [2]. Therefore, it would be interesting to extend the model to other community populations, also including behavioural variables to understand their effect on lifestyles, exercise and nutrition regarding the change in MetS status change [5,68]. In addition, future studies could explore the relationship between childhood and adulthood in the change in MetS status [40,41,42,43,44,45]. Multivariate models may support clinical decision-making associated with health promotion to develop mitigation and management strategies for this major public health concern [82].

## 5. Conclusions

Current research provides the first adjusted goodness-of-fit model for predicting the relative contribution of each component in the change in MetS status, specifically in Portuguese adults from a north-eastern Portuguese region. The multivariate model confirmed the highest relative contribution of WC, IFG and SBP during changes in MetS, whether 3-, 4- or 5-component. Additionally, an indirect effect was reported for age and body composition in MetS status change. The absence and/or trivial direct effect reported for DBP, TG and HDL in MetS evolution may suggest that other components with higher specificity and sensitivity should be considered to empirically validate the harmonised definition of MetS. Futures research should add independent variables into the multivariate model to prescribe and control the exercise intensity and volume in the change in MetS status. 

## Figures and Tables

**Figure 1 ijerph-19-03384-f001:**
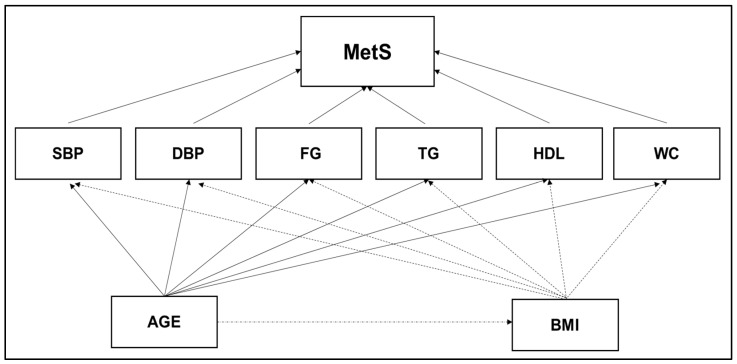
Theoretical model designed according to JIS criteria to measure the direct effect for each MetS component and the indirect effect for age and body composition. Abbreviations: BMI—body mass index; DBP—diastolic blood pressure; FG—fasting glucose; TG—triglycerides; HDL—low high-density lipoprotein cholesterol; SBP—systolic blood pressure; WC—waist circumference.

**Figure 2 ijerph-19-03384-f002:**
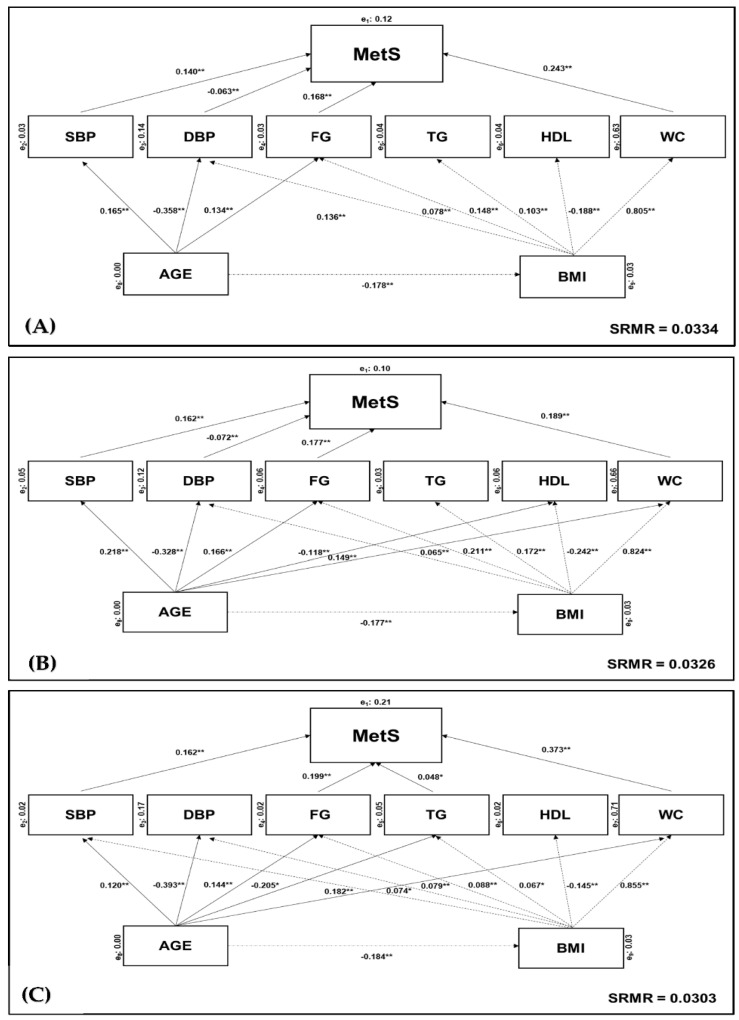
Adjusted goodness-of-fit model for the effects of MetS components. Analysed according to: (**A**) overall Portuguese population (***SRMR*** = 0.0334); (**B**) female sub-group only (***SRMR*** = 0.0326); (**C**) male sub-group only (***SRMR*** = 0.303). Endogenous (*y_i_*) variable depends on exogenous (*x_i_*) variable (*x_i_* → *y_i_*). Significant differences were verified as: ** *p* < 0.001; * *p* < 0.05. Abbreviations: BMI—body mass index; DSP—diastolic blood pressure; FG—fasting glucose; *p*—level of significance (*p* value); HDL-c—low high-density lipoprotein cholesterol; MetS—metabolic syndrome; SBP—systolic blood pressure; SMRM—standardised root mean square residuals; TG—triglycerides; WC—waist circumference.

**Table 1 ijerph-19-03384-t001:** Descriptive and frequencies for MetS components according to sex and overall population.

Variables	Women (*n* = 1914)	Men (*n* = 1667)	Total (*n* = 3581)
Age (year) [M ± SD]	64.08 ± 14.23	66.73 ± 12.89	65.50 ± 13.59
18–39 years [*n* (%)]	55 (2.9)	102 (6.1)	157 (4.4)
40–64 years [*n* (%)]	696 (36.4)	642 (38.5)	1338 (37.4)
>65 years [*n* (%)]	1163 (60.8)	923 (55.4)	2086 (58.3)
Height (cm) [M ± SD]	157.02 ± 6.16	169.22 ± 6.84	162.70 ± 5.18
Weight (kg) [M ± SD]	71.55 ± 13.89	83.56 ± 13.37	77.14 ± 14.91
BMI (kg/cm^2^) [M ± SD]	28.98 ± 5.18	29.135 ± 4.00	29.05 ± 4.67
Normal [*n* (%)]	426 (22.3)	237 (14.2)	663 (18.5)
Overweight [*n* (%)]	776 (40.5)	771 (46.3)	1547 (43.2)
Obesity [*n* (%)]	712 (37.2)	659 (39.5)	1371 (38.3)
3-MetS components [*n* (%)]	729 (38.1)	630 (37.8)	1359 (38.2)
4-MetS components [*n* (%)]	651 (34.0)	639 (38.3)	1290 (36.0)
5-MetS components [*n* (%)]	534 (27.9)	398 (23.9)	932 (26.0)
SBP (mg/dL) [M ± SD]	134.99 ± 14.02	137.33 ± 13.68	136.08 ± 13.68
↑ SBP (mmHg) [*n* (%)]	1308 (68.3)	1254 (75.2)	2562 (71.5)
DBP (mg/dL) [M ± SD]	75.37 ± 9.64	76.04 ± 10.23	75.69 ± 9.92
↑ DBP (mmHg) [*n* (%)]	326 (17.0)	339 (20.3)	1254 (75.2)
FG (mg/dL) [M ± SD]	107.35 ± 28.18	116.58 ± 35.23	111.65 ± 31.99
IFG (mg/dL) [*n* (%)]	1037 (54.2)	1136 (68.1)	1254 (75.2)
TG (mg/dL) [M ± SD]	121.04 ± 58.41	130.29 ± 13.68	125.35 ± 78.85
↑ TG (mg/dL) [*n* (%)]	475 (24.8)	463 (27.8)	1254 (75.2)
HDL (mg/dL) [M ± SD]	59.39 ± 14.12	51.20 ± 12.77	55.58 ± 14.11
↑ HDL (mg/dL) [*n* (%)]	120 (6.3)	845 (50.7)	1254 (75.2)
WC (cm) [M ± SD]	97.69 ± 11.79	103.67 ± 10.16	100.47 ± 11.46
↑ WC (cm) [*n* (%)]	1595 (83.3)	946 (56.7)	1254 (75.2)

Abbreviations: ↑—elevated; BMI—body mass index; DBP—diastolic blood pressure; FG—fasting glucose; HDL—low high-density lipoprotein cholesterol; IFG—impaired fasting glucose; SBP—systolic blood pressure; TG—triglycerides; WC—waist circumference.

**Table 2 ijerph-19-03384-t002:** Mean comparison for MetS components according to sex group, BMI bands and interaction effect for age group × BMI bands.

		Age Group				BMI				Age Group × BMI		
Variables	M ± SD	*F*	*p*	η^2^	Pairwise	*F*	*p*	η^2^	Pairwise	*F*	*p*	η^2^
**Women (*n* = 1914)**												
SBP (mmHg)	134.99 ± 14.02	34.923	0.000	0.035	a, b, c	0.971	0.379	0.001	–	0.728	0.573	0.002
DBP (mmHg)	75.37 ± 9.64	75.807	0.000	0.074	b, c	1.793	0.167	0.002	–	0.756	0.554	0.002
FG (mg/dL)	107.35 ± 28.18	18.647	0.000	0.019	b, c	14.067	0.000	0.015	a, b, c	1.298	0.269	0.003
TG (mg/dL)	121.04 ± 58.41	14.066	0.000	0.015	b, c	0.856	0.425	0.001	–	2.890	0.021	0.006
HDL (mg/dL)	59.39 ± 14.12	4.821	0.008	0.005	b	16.347	0.000	0.017	a, b, c	0.512	0.727	0.001
WC (cm)	97.69 ± 11.79	17.138	0.000	0.018	–	262.754	0.000	0.216	–	1.213	0.303	0.003
**Men (*n* = 1667)**												
SBP (mmHg)	137.33 ± 13.68	9.001	0.000	0.011	b, c	1.142	0.319	0.001	–	1.606	0.170	0.004
DBP (mmHg)	76.04 ± 10.23	122.625	0.000	0.129	b, c	0.838	0.433	0.001	–	1.782	0.130	0.004
FG (mg/dL)	116.58 ± 35.23	7.696	0.000	0.009	a, b	0.297	0.743	0.000	–	1.991	0.093	0.005
TG (mg/dL)	130.29 ± 13.68	20.535	0.000	0.024	b, c	3.785	0.023	0.005	b, c	0.990	0.412	0.002
HDL (mg/dL)	51.20 ± 12.77	3.995	0.019	0.005	a, b	6.109	0.002	0.007	b, c	0.188	0.945	0.000
WC (cm)	103.67 ± 10.16	26.481	0.000	0.031	b	408.137	0.000	0.330	c	0.253	0.908	0.001
**Total (*n* = 3581)**												
SBP (mmHg)	136.08 ± 13.68	37.574	0.000	0.021	a, b, c	2.444	0.087	0.001	–	1.654	0.158	0.002
DBP (mmHg)	75.69 ± 9.92	199.313	0.000	0.100	b, c	0.970	0.379	0.001	–	2.245	0.062	0.003
FG (mg/dL)	111.65 ± 31.99	21.831	0.000	0.012	a, b, c	5.945	0.003	0.003	a, b, c	1.446	0.216	0.002
TG (mg/dL)	125.35 ± 78.85	27.960	0.000	0.015	a, b, c	5.412	0.004	0.003	a, b, c	1.584	0.176	0.002
HDL (mg/dL)	55.58 ± 14.11	5.088	0.006	0.003	a, c	27.302	0.000	0.015	a, b, c	0.612	0.654	0.001
WC (cm)	100.47 ± 11.46	30.988	0.000	0.017	a	610.302	0.000	0.255	a, b, c	0.758	0.552	0.001

Significant differences were verified for Turkey’s post hoc according to age group and BMI bands: (a) Young adults (18–39 years) vs. middle-age adults (40–64 years); (b) young adults (18–39 years) vs. older adults (>65 years); (c) middle-age adults (40–64 years) vs. older adults (>65 years). *Abbreviations*: BMI—body mass index; DBP—diastolic blood pressure; FG—fasting glucose; HDL—low high-density lipoprotein cholesterol; SBP—systolic blood pressure; TG—triglycerides; WC—waist circumference.

## Data Availability

Data are available upon request to the contact author.

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
