# Peer review of "Structural Equation Modelling for Predicting the Relative Contribution of Each Component in the Metabolic Syndrome Status Change"

_ijerph, 2022, doi:10.3390/ijerph19063384_

Round 1

Reviewer 1 Report

The study by Teixeira and colleagues aimed to develop a confirmatory model to describe and explain the direct and indirect effect of each component in MetS status change. I have some comments as follows:

1. The variable such as TG is usually supposed to be not normally distributed. However, the characteristics of the TG were shown as mean and standard deviation, not median and interquartile range. What were the results of the Kolmogorov–Smirnov test?  If the TG variable was not normally distributed, it was shown as median and interquartile range. 

2. Language, text, and grammar

There is a need to go through the manuscript and improve the language, text, and grammar. Some typographic and orthographic errors should be corrected throughout the whole manuscript.

ex) line 83: specifially->specifically, Line 90: Splited-> spilt, line 115: centre->centres, line 171~172

3. Structural equation model (line 206~218)

The significant digits of β are different.

Author Response

Thank you very much for the time you spent and your feedback on this manuscript. We have made every effort to take on board your recommendations and comments. We hope this revised version and the responses to the comments (kindly refer to our replies below) will meet your requirements. Also, an English language revision was conducted to improve the language quality of the original article. Please note that all new changes in the revised manuscript are edited with the Microsoft Word® tracking tool.

Please, see attachment.

Reviewer 2 Report

  • Please mention the date of sampling took place in the abstract.
  • Line 27-29: The conclusion should state the wider implications of your findings.
  • Line 39-47: It would be good to justify what has been found in other European countries and why it is important to focus on the Portuguese context.
  • Line 70-79: Please specify where previous research discussed has been conducted.
  • Line 80-81: There is weak evidence describing the validation of MetS score by confirmatory factor analysis in children and adults. I would recommend referring to these articles (Diabetologia. 2014;57(5):940-9; Cardiovasc Diabetol. 2012; 11: 128; Diabetes Care. 2010;33(6):1370-2; Lipids in Health and Disease. 2013,12, 61; PLoS. 2018; 13(12), e0208231).
  • Line 81-85: Need to include a stronger rationale for looking at individuals with MetS diagnosis aged 18-102 years. A justification for why WC, SBP and IFG achieved a higher relative contribution for predicting the MetS status change at this age group. How does evidence in adults relate to childhood?
  • Line 88-92: Inclusion and exclusion criteria should be clearly defined.
  • Line 92-95: What’s the theoretical basis for stratifying the models by sex? There is no hypothesized to differ between men and women.
  • Line 113-119: Additional information is required of all laboratory measures. Data collection procedure should be described in sufficient details.
  • The details (numerical results) in figure 2 should be clear enough to the reader.
  • Line 242-245; line 275-280: Please do not repeat results in the discussion.
  • I miss a discussion on MetS and its components according to sex.

Author Response

(The authors gave the same response as above.)

Reviewer 3 Report

Thank you for the opportunity to contribute to the peer review process for the original study submission manuscript entitled “Structural Equation Modelling for Predicting the Relative Contribution of each Component in the Metabolic Syndrome Status Change”. The manuscript is interesting, well written and points out relevant issues.

My comments:

2.1. Study design and population

  • Where are the individuals come from? I saw in the acknowledgments that there were two primary health care centers. Please, describe these centers comparing to Portugal or the city where the study was conducted.
  • When did you do the data collection? Please include.

2.5. Statistical analysis

            - The post-hoc test used in Table 2 should be cited.

Table 1 -> Exclude the line about MetS[n (%)], as the n is already stated in header and one can easily calculate the %. I suggest this because the table was designed with column % and only this line is about the row %.

Table 1 -> Please amend the numbers for Total column from ­SBP to the end.

2.2.1. Anthropometric measures => please clarify who (physicians, students, helpers of any kind) and how many different researchers participated in this process.

2.2.2. Laboratory analysis and blood pressure – same question above for SBP and DBP

Figure 2 – BMI and HDL were inversely related. I think this should be cited and discussed.

Line 230 – I am not sure your sample is “a community-representative sample of Portuguese adults”. Please clarify this in the Methods

Line 279 – Clarify the sentence: “every effect is related to HDL on MetS evolution for sampled population”

Line 325 – this should be in the Methods section

Author Response

(The authors gave the same response as above.)

Round 2

Reviewer 2 Report

Thanks to the authors for the revised version of the paper. It has clearly improved. One point remains

Line 84-85: Please rephrase this statement.  Authors should add more details on describing these studies [40–45], and clearly state why this work is important in light of these studies?

Author Response

Thank you very much for the time you spent and your feedback in this second round of revisions. We have made every effort to take on board your minor revisions. We hope this second revised version and the responses to the comment (kindly refer to our replies below) will meet your requirements. Please, note that the revision from this second round is edited with the Microsoft Word® tracking tool.

Please, see attachment.
